# Glycogen engineering improves the starvation resistance of mesenchymal stem cells and their therapeutic efficacy in pulmonary fibrosis

Yongyue Xu[1,2†], Mamatali Rahman[1,2,3†], Zhaoyan Wang[1,2], Bo Zhang[1,2], Hanqi Xie[1,2], Lei Wang[1,2], Haowei Xu[1,2], Xiaodan Sun[4], Shan Cheng[5], Qiong Wu[1,2,6*]

[1]MOE Key Laboratory of Bioinformatics, Center for Synthetic and Systems Biology, Tsinghua University, Beijing, China; [2]School of Life Sciences, Tsinghua University, Beijing, China; [3]Xinjiang Stem Cells Special Plateau Disease Engineering Technology Research Center, Kashi, China; [4]School of Materials Science and Engineering, Tsinghua University, Beijing, China; [5]Department of Medical Genetics and Developmental Biology, School of Basic Medical Sciences, Capital Medical University, Beijing, China; [6]State Key Laboratory of Green Biomanufacturing, Beijing, China

*For correspondence:
wuqiong@mail.tsinghua.edu.cn

†These authors contributed equally to this work

Competing interest: The authors declare that no competing interests exist.

## eLife Assessment

This **important** study presents a novel approach to enhance the therapeutic potential of mesenchymal stromal cells (MSCs) by genetically modifying their glycogen synthesis pathway, resulting in increased glycogen accumulation and improved cell survival under starvation conditions, particularly in the context of experimental pulmonary fibrosis. The methods and findings are generally **solid** and could be strengthened in the future by investigating the kinetics of persistence, the immunomodulatory effects, and the underlying improved mechanism of action of MSCs in this pulmonary fibrosis model. If confirmed, this approach could suggest potential methods to improve the therapeutic functionality of MSCs in cell therapy strategies.

**Abstract** Mesenchymal stem cells (MSCs) are widely used in regenerative medicine, including the treatment of pulmonary fibrosis. However, implanted MSCs disappear within days, constraining therapeutic efficacy, which is largely attributed to nutrient deprivation. In this study, we established glycogen metabolism engineering strategies in mammalian cells. By expressing a functionally optimized glycogen synthase (GYSmut), MSCs could accumulate large amounts of glycogen rapidly as a reserve substance. Glycogen engineering significantly improved the survival of MSCs during starvation both in vitro and in vivo, enhancing cell viability post-implantation and their therapeutic efficacy in pulmonary fibrosis. Glycogen-engineered MSCs may serve as chassis cells for further applications. Our research highlights the importance of glucose metabolism regulation in cell-based therapy and demonstrates the great potential for the metabolic engineering of MSCs and other therapeutic cells.

## Introduction

Pulmonary fibrosis (PF) is traditionally regarded as an irreversible lung disease for which there are generally no effective drugs, with the exception of nintedanib and pirfenidone (*Alonso-Gonzalez et al., 2023*; *Bari et al., 2021*). Regenerative medicine represents an alternative approach, with several

successful therapeutic applications in patients with chronic lung diseases, including PF (*Li et al., 2020*; *Li et al., 2021*; *Cheng et al., 2017*). The recent focus on mesenchymal stem cells (MSCs) has generated enthusiasm due to their unique regenerative and immunomodulating properties, including the ability to reverse fibrosis (*Rahman et al., 2022*; *Zhao et al., 2023*). Recent studies have begun to unravel the mechanisms underlying the therapeutic potential of MSCs, with a particular emphasis on their secretory properties. The paracrine factors secreted by MSCs, including fibroblast growth factor and cytokines, have been shown to mediate tissue repair and regeneration (*Rahman et al., 2022*; *Tian et al., 2023*). However, the therapeutic efficacy of MSCs has been limited due to their poor survival and short dwell time in the hostile pathological microenvironment, which has hindered the translation of these promising preclinical findings into clinical practice (*Moya et al., 2018*; *Levy et al., 2020*).

Metabolic regulation is crucial for cell survival post-implantation. It has been reported that the hypoxic in vivo microenvironment inhibits oxidative phosphorylation in implanted MSCs, forcing them to rely on glycolysis instead, which yields less ATP (*Ladurner, 2006*; *Lavrentieva et al., 2010*; *Das et al., 2010*). Our previous single-cell transcriptomic study also showed that the glucose metabolism pathway of MSCs was downregulated in the PF microenvironment (*Rahman et al., 2022*). Compared

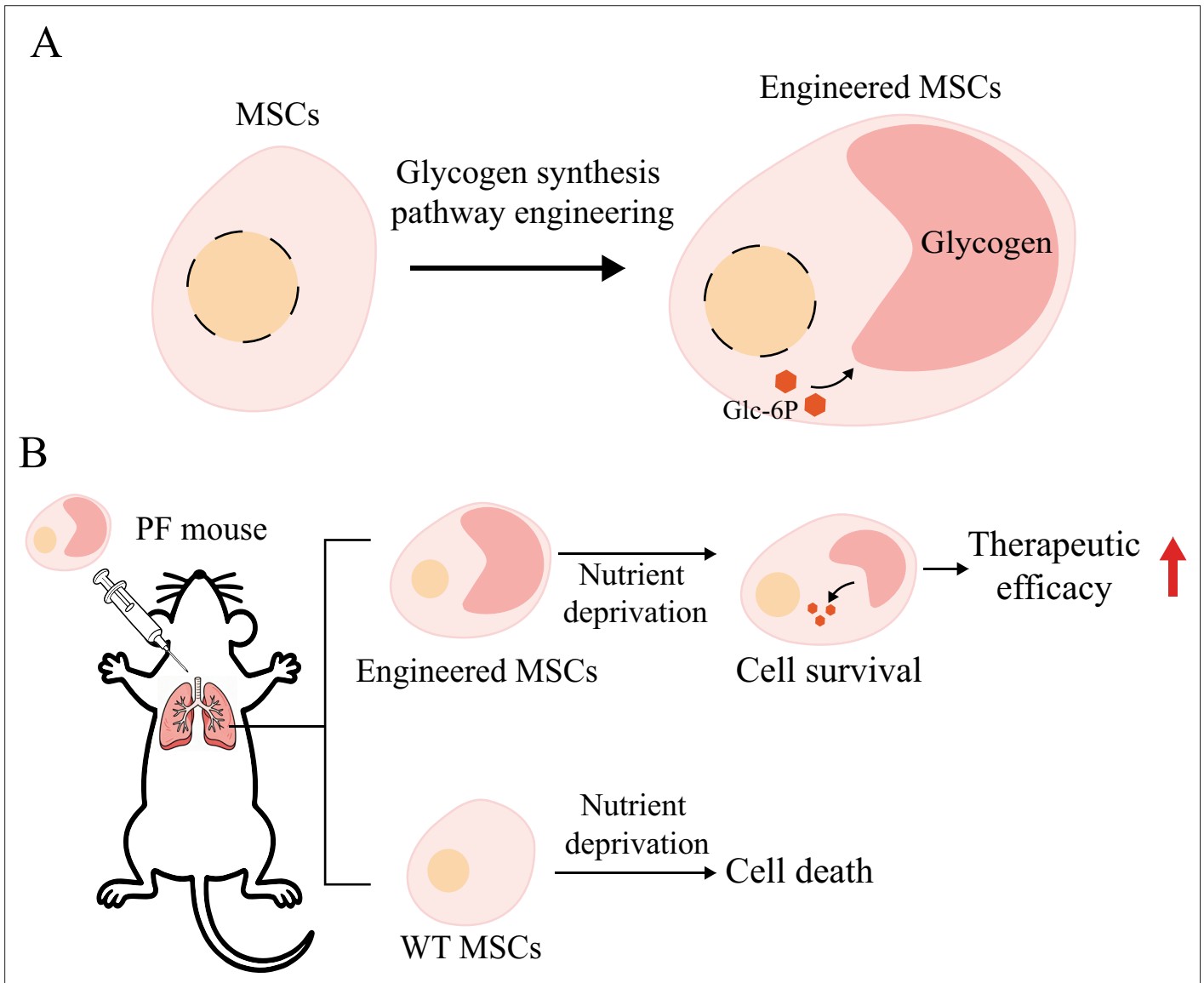

**Figure 1.** Schematic illustration of the strategy used to enhance the therapeutic efficacy of mesenchymal stem cells (MSCs) through glycogen engineering. (**A**) Engineering glycogen metabolism of MSCs by overexpressing essential enzymes of glycogen synthesis. (**B**) Engineered MSCs use glycogen as an energy supply after implantation, improving cell viability and therapeutic efficacy.

with in vitro culture systems, it is more difficult for implanted MSCs to obtain sufficient glucose in pathological microenvironments, and recent findings suggest that glucose exhaustion is the main reason for cell death after implantation (*Salazar-Noratto et al., 2020*). Supplying glucose to MSCs improved their survival post-implantation (*Deschepper et al., 2013*). Therefore, glucose metabolism engineering has the potential to improve MSC therapy by enhancing starvation resistance and prolonging the residence time of implanted cells.

Glycogen is a branched polymer that acts as the primary glucose storage form in mammalian cells, supplying energy under glucose deficiency and promoting cell survival. Glycogen metabolism is controlled by a series of enzymes, including glycogen synthase (GYS) (*Roach et al., 2012*). To reduce cell death of implanted MSCs caused by nutrient deprivation, we engineered the glycogen metabolism of MSCs with genes encoding essential enzymes, which promoted glycogen accumulation (*Figure 1A*). The accumulated glycogen serves as an energy supply post-implantation, improving the viability and therapeutic efficacy of MSCs (*Figure 1B*).

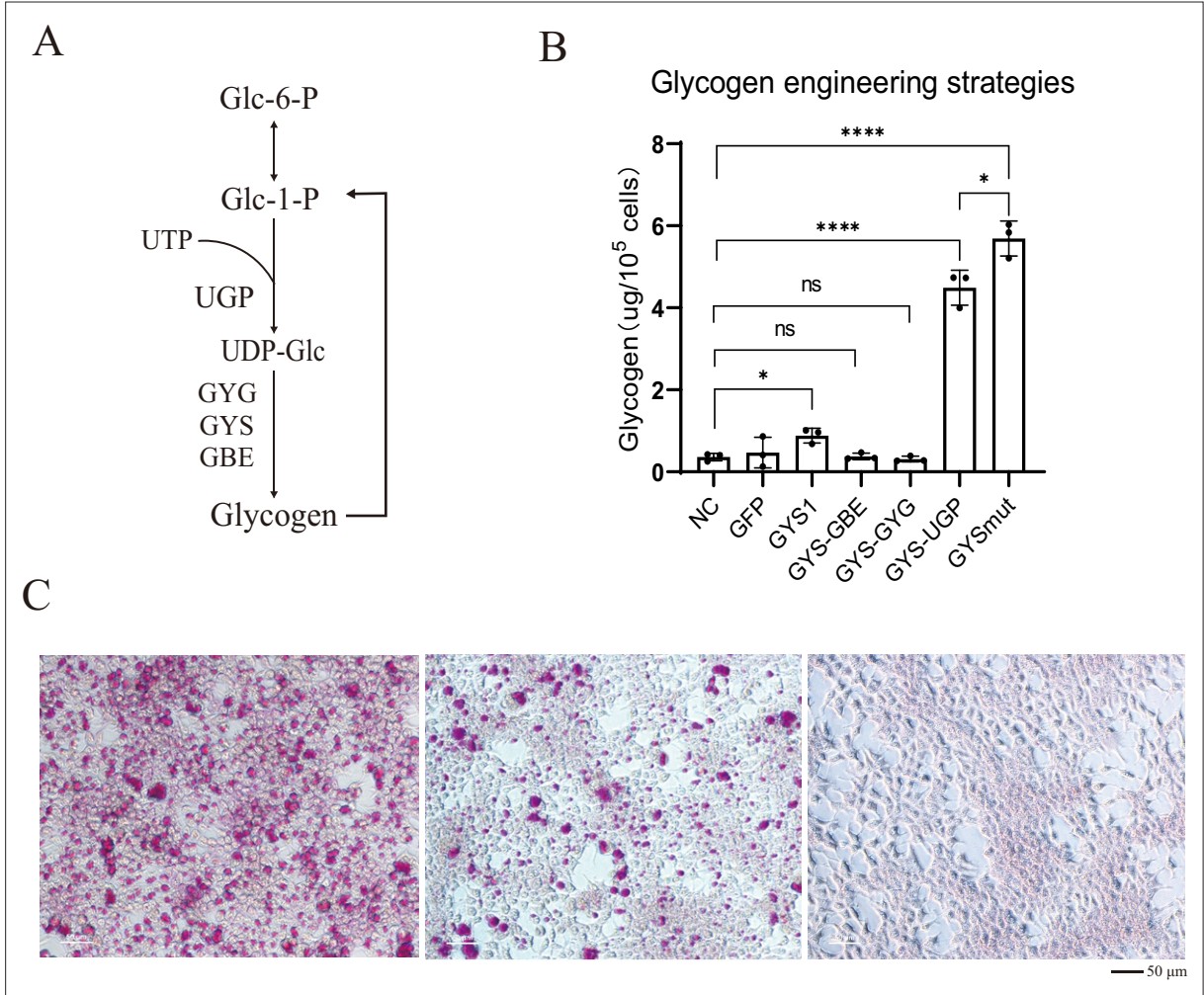

**Figure 2.** Construction of glycogen engineering strategies. (**A**) Essential enzymes of glycogen synthesis. (**B**) Tests of essential enzyme combination strategies in HEK293T cells by transient transfection. The glycogen content of each group was measured (N=3, mean ± SD). (**C**) Periodic acid-Schiff (PAS) staining of cells expressing GYSmut and glycogen synthase (GYS)-UDP-glucose pyrophosphorylase (UGP) revealed significant accumulation of glycogen granules.Scale bar, 50 μm.

The online version of this article includes the following figure supplement(s) for figure 2:

**Figure supplement 1.** Further optimization of glycogen engineering strategies.

## Results

### Construction of glycogen strategies in mammalian cells

Glycogen is synthesized from UDP-glucose in a process involving glycogenin (GYG), UDP-glucose pyrophosphorylase (UGP), GYS, and glycogen-branching enzyme (GBE) (*Roach et al., 2012*). GYG is the initiator of glycogen synthesis, UGP catalyzes the synthesis of UDP-glucose, GYS is responsible for the synthesis of glycogen, and GBE introduces branches into the structure of glycogen (*Figure 2A*). Here, we first devised glycogen engineering strategies based on these essential enzymes.

Initially, we transiently expressed *GYS1* in human HEK293T cells, but it only had a limited, albeit significant effect on glycogen accumulation (*Figure 2B*). We attributed the unsatisfactory results to substrate limitation, as well as potential inhibition of *GYS1* by cellular pathways. Next, we tried different expression strategies to increase glycogen accumulation, including (1) expressing *GYS1* alone to promote glycogen synthesis, (2) co-expressing *GYS1* with *GBE1* to increase glycogen branching, (3) co-expressing *GYS1* with *GYG1* to increase glycogen initiation, and (4) co-expressing *GYS1* with *UGP2* to increase the substrate supply. Some protein kinases (e.g. AMPK) inactivate GYS1 by phosphorylation at specific serine sites (*Jørgensen et al., 2004*). Therefore, (5) we introduced Ser-Ala mutations at the Ser-8, Ser-641, Ser-645, Ser-649, Ser-653, Ser-654, and Ser-657 sites of GYS1 (GYSmut) to prevent phosphorylation (see sequences in *Supplementary file 1*). Both co-expression of *GYS1* with *UGP2* and expression of GYSmut alone resulted in greatly increased glycogen accumulation compared to the control (~12- and 16-folds) and other strategies (*Figure 2B*). Glycogen granules were clearly visible following periodic acid-Schiff (PAS) staining (*Figure 2C*).

Since most of the Ser-Ala mutation sites were located near the C-terminus of GYSmut, we truncated the 100 C-terminal amino acids to generate GYSdelc. Although this truncated variant was still capable of inducing cellular glycogen accumulation, it exhibited a reduced rate in comparison to full-length GYSmut (*Figure 2—figure supplement 1A*). Further, we co-expressed GYSmut with UGP, which demonstrated a marked additive effect (*Figure 2—figure supplement 1B*).

### Glycogen accumulation improved the starvation resistance of MSCs

After expressing GYSmut in MSCs by lentivirus transduction, glycogen accumulation was significantly increased in the engineered MSCs (*Figure 3A*). PAS staining showed more PAS-positive glycogen granules in GYSmut MSCs (*Figure 3B*), which altered cell morphology by significantly increasing the cell volume, as assessed by flow cytometry (*Figure 3—figure supplement 1A*). Glycogen granules were located in both the cytoplasm and the nucleus (*Figure 3—figure supplement 1B*). Interestingly, when co-overexpressing wild-type GYS1 and GYG, we observed a diffuse intracellular glycogen distribution rather than bulky glycogen particles, indicating that the combination of distinct enzymes influences the characteristics of glycogen synthesis (*Figure 3—figure supplement 1C*).

To evaluate the starvation resistance of glycogen-engineered MSCs, we cultured GYSmut MSCs in Dulbecco's phosphate-buffered saline (DPBS) instead of a complete culture medium to create a nutrient-deprived environment. The Cell Counting Kit-8 assay was used to measure cell viability at various time points. Notably, approximately 40% of the GYSmut MSCs were still viable after 72 hr of starvation culture, whereas wild-type MSCs died rapidly after 8 hr (*Figure 3C*). After 48 hr of starvation, GYSmut MSCs still retained a 39% glycogen reserve (*Figure 3D*), indicating that the accumulated glycogen could be a long-term energy resource to maintain higher cell viability during nutrient deprivation following implantation. Since some pathological contexts are hypoxic, we tested the starvation resistance of MSCs under hypoxia (oxygen concentration <5%), with the engineered MSCs still maintaining a higher survival proportion (*Figure 3—figure supplement 1D*).

As glycogen is a crucial node of carbon metabolism in mammalian cells, we next investigated whether glycogen engineering caused global changes in MSCs. Cell counting showed that glycogen engineering had no significant effect on cell proliferation (*Figure 3E*). To evaluate the effects of glycogen engineering on the adipogenic differentiation capacity of MSCs, we induced GYSmut MSCs with adipocyte medium for 5 days, followed by Oil Red O staining and qRT-PCR of the adipocyte differentiation marker *Lpl*. The results indicated a decrease in the adipocyte differentiation potential of GYSmut MSCs (*Figure 3F and G*). It was reported that adipogenic differentiation and the immunomodulatory functions of MSCs are mutually exclusive (*Li et al., 2024*), so blocking adipocyte differentiation may enhance the immunosuppressive activity of MSCs.

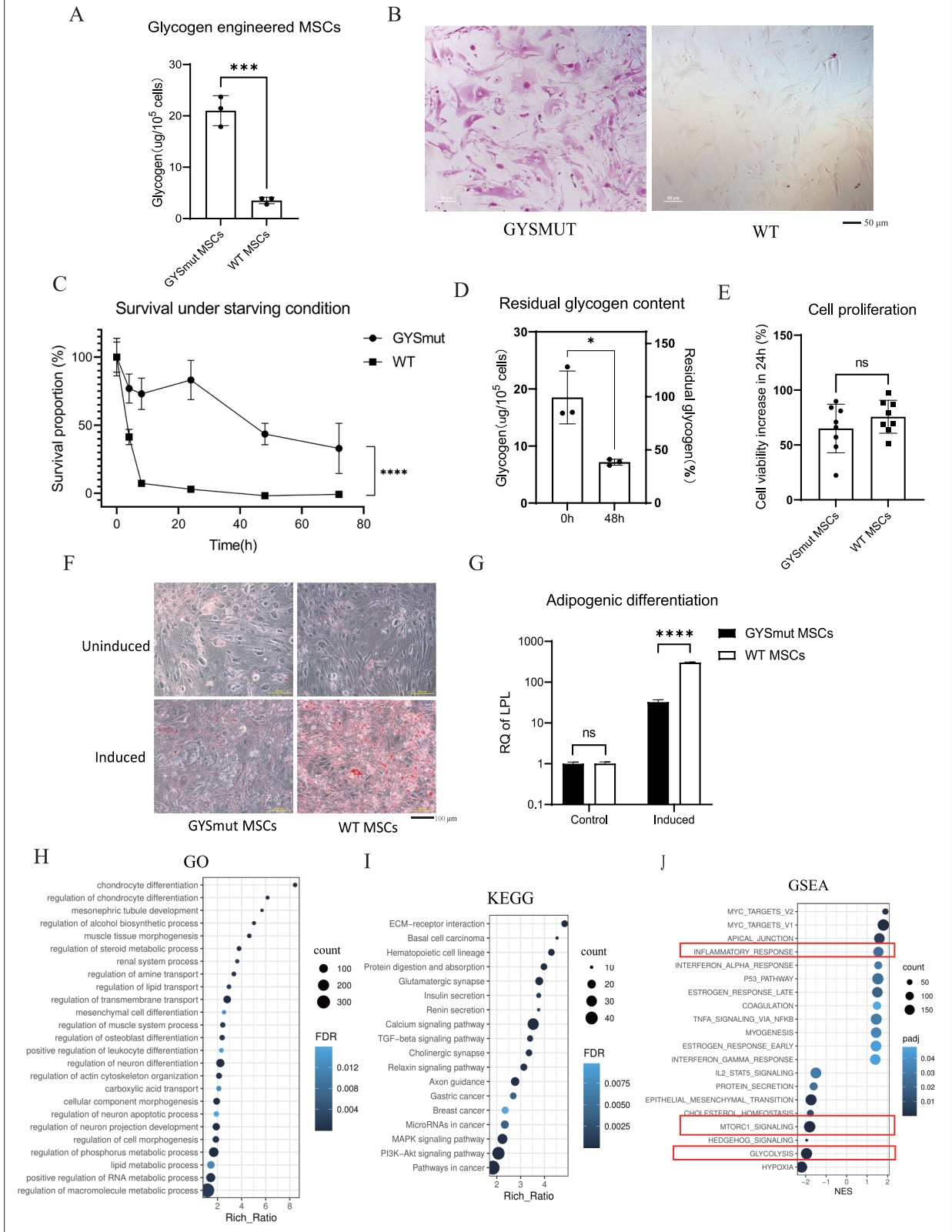

**Figure 3.** Construction of glycogen-engineered mesenchymal stem cells (MSCs). (**A**) Glycogen content of GYSmut MSCs (N=3, mean ± SD). (**B**) Periodic acid-Schiff (PAS) staining of GYSmut MSCs, showing glycogen granules (red). Scale bar, 50 μm. (**C**) Survival of engineered MSCs under Dulbecco's phosphate-buffered saline (DPBS) (starvation) treatment in vitro (N=3, mean ± SD). (**D**) Residual glycogen content of GYSmut MSCs after DPBS (starvation) treatment for 48 hr (N=3, mean ± SD). (**E**) Viability of GYSmut MSCs according to the CCK8 assay (N=8, mean ± SD). (**F, G**) Adipogenic

*Figure 3 continued on next page*

*Figure 3 continued*

differentiation potential of GYSmut MSCs, assessed by Oil Red O staining and qPCR detection of Lpl expression (N=3, mean ± SD, unpaired *t*-test p-value<0.0001). Scale bar, 100 µm. (**H**) Gene ontology (GO) enrichment analysis of differentially expressed genes (DEGs) between GYSmut MSCs and the GFP control. (**I**) KEGG analysis of DEGs. (**J**) Gene set enrichment analysis (GSEA) of the DEGs.

The online version of this article includes the following figure supplement(s) for figure 3:

**Figure supplement 1.** Impacts of glycogen engineering on mesenchymal stem cells (MSCs).

We next performed bulk RNA sequencing (RNA-seq) to understand the effect of glycogen engineering on the transcriptome of GYSmut MSCs. A total of 759 genes were upregulated and 542 were downregulated in GYSmut MSCs, indicating a broad effect on the transcriptome (*Figure 3—figure supplement 1E*). Gene ontology enrichment analysis revealed that differentially transcribed genes were predominantly enriched in processes related to cellular differentiation, morphogenesis, and the regulation of various metabolic pathways, including lipid metabolism, reflecting potential metabolic reprogramming. The differentially expressed genes were also enriched in the terms related to processes of muscle, kidneys, and neural tissue, which are sites where glycogen is naturally distributed in mammals (*Figure 3H*; *Roach et al., 2012*). KEGG analysis showed the PI3K-Akt signaling pathway was also modulated, which is crucial for cell survival (*Figure 3I*). The gene set enrichment analysis (GSEA) results showed that the inflammatory response gene set was upregulated, which may be beneficial in the treatment of PF, while mTORC1 and glycolysis-related gene sets were downregulated (*Figure 3J*). It was reported that mTORC1 responds to cellular amino acid and glucose levels (*Szwed et al., 2021*), which is in agreement with the hypothesis that glycogen synthesis affects cellular glucose levels and the process of glycolysis.

## Glycogen engineering promoted the survival of MSCs post-implantation

In our previous study, intratracheal administration of MSCs exhibited a robust therapeutic effect in a mouse model of bleomycin-induced PF (*Rahman et al., 2022*), and further scRNA-seq analysis of implanted MSCs within the PF microenvironment revealed downregulation of glucose metabolism (*Figure 4—figure supplement 1*), suggesting that nutrient availability is a critical determinant of MSC viability and may potentially limit their therapeutic efficacy. We, therefore, tested whether glycogen engineering would promote the survival of GYSmut MSCs in vivo. PF mice were intratracheally injected with glycogen-engineered MSCs co-expressing GYSmut and *Gaussia* luciferase (GYSmut-Gluc), while the control group was injected with Gluc MSCs. We collected lungs of PF mice on days 0 and 7 after MSC implantation and measured the fluorescence intensity of lung tissue homogenates with a commercial kit to assess the counts of living MSCs (three mice each group each time point) (*Figure 4A*). A higher proportion of surviving GYSmut-Gluc MSCs was detected compared with the control (*Figure 4B*).

Because of the limited sensitivity of *Gaussia* luciferase labeling for in vivo imaging, to assess the persistence and biodistribution of implanted MSCs within the PF microenvironment, we used a different luciferase, *Akaluc* luciferase. GYSmut MSCs co-expressing Akaluc (GYSmut-Akaluc) were intratracheally injected into PF mice. This approach, in combination with the substrate Akalumine (*Bozec et al., 2020*), allows for highly sensitive live imaging and thus provides insights into the retention of implanted cells. In the control group, MSCs expressing Akaluc were injected (five mice each group). As expected, on day 7 post-implantation, higher fluorescence intensity was detected in the GYSmut MSCs group compared with the control (*Figure 4C*), indicating the greater survival ability of GYSmut MSCs compared to wild-type MSCs in the PF microenvironment. Quantification of fluorescent imaging revealed a further decline in survival rates to similar levels in both GYSmut MSCs and the control by day 11, suggesting that glycogen engineering appears to be primarily functional during the first 2 weeks post-implantation (*Figure 4—figure supplement 2*).

## Engineered MSCs demonstrated a beneficial effect on PF

To investigate the therapeutic efficacy of glycogen-engineered MSCs (GYSmut MSCs), we administered them via tracheal injection into mice on day 3 following bleomycin treatment and recorded the survival rate and body weight of mice over time (10 mice each group). GYSmut MSCs demonstrated

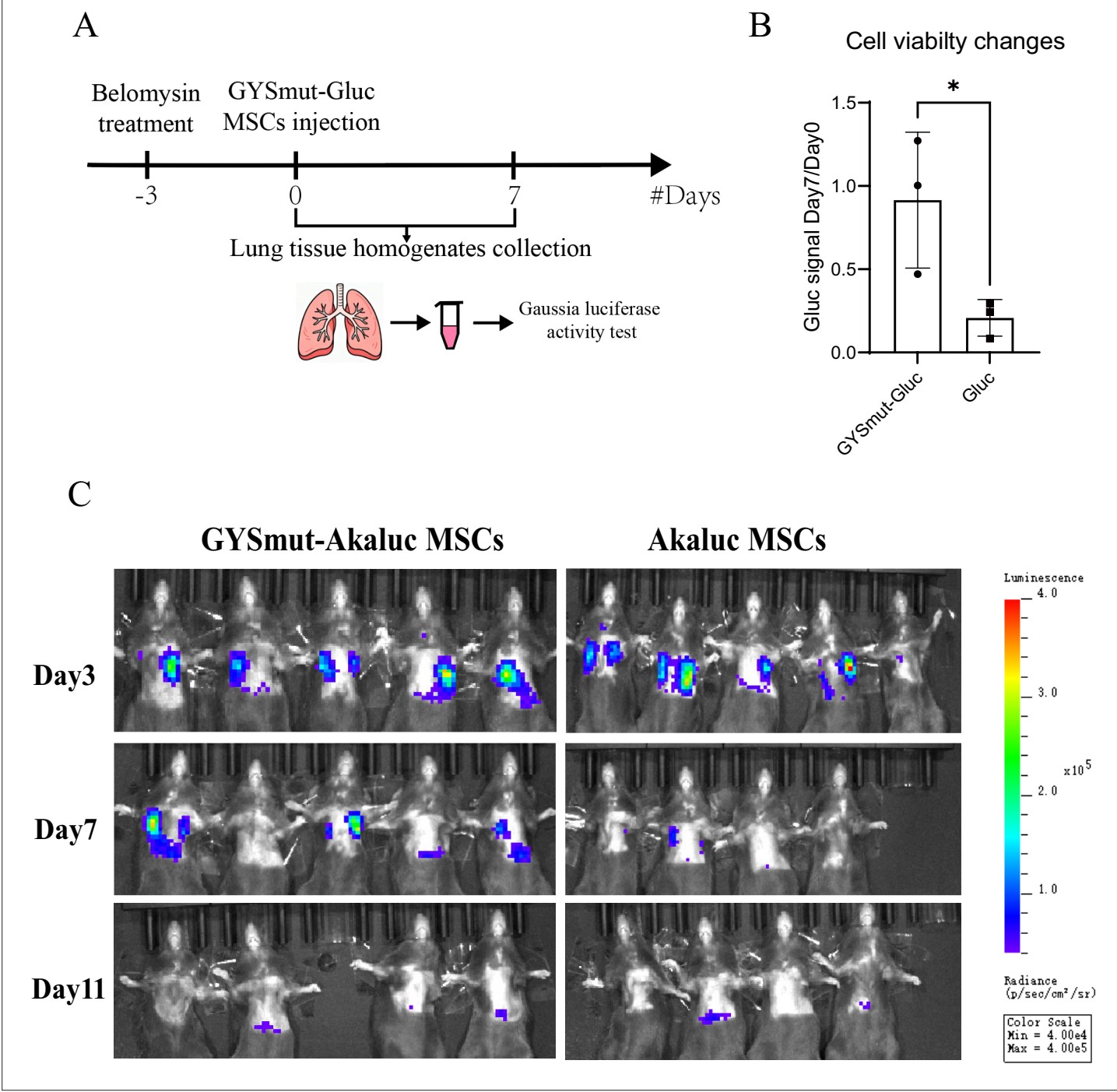

**Figure 4.** Survival of implanted glycogen-engineered mesenchymal stem cells (MSCs). (**A**) Schematic illustration of the strategy used to assess the survival of MSCs by detecting *Gaussia* luciferase activity in the homogenate of lungs. (**B**) Changes of *Gaussia* luciferase activity in the two groups on day 7 post-implantation (three mice each group, unpaired *t*-test p-value = 0.044) (N=3, mean ± SD). (**C**) Live imaging of *Akaluc* luciferase activity in the two groups implanted with GYSmut-Akaluc MSCs and control cells (five mice each group). One mouse from the control group died on day 7, and one mouse from the GYSmut-Akaluc group died on day 11.

The online version of this article includes the following figure supplement(s) for figure 4:

**Figure supplement 1.** KEGG and gene set enrichment analysis (GSEA) of implanted mesenchymal stem cells (MSCs) in our previous research.

**Figure supplement 2.** Quantification of Akaluc activity of in vivo imaging post-implantation.

a more pronounced effect than the control group expressing only GFP (*Figure 5A*). Quantitative analysis of lung tissue sections stained with hematoxylin and eosin (H&E) and Masson's trichrome revealed that the GYSmut MSCs group exhibited superior therapeutic efficacy on day 7, as evidenced by reduced collagen deposition and preserved alveolar size (quantified by mean linear intercept [MLI]) compared to the control (*Figure 5B and C*). The improved therapeutic efficacy illustrated the beneficial effect of enhanced MSC viability and confirmed the rationale for glycogen engineering.

## Discussion

Glycogen is found in a wide range of organisms, ranging from bacteria to mammals. As a major form of stored glucose, glycogen plays an essential role in maintaining energy homeostasis and cell survival. In this study, we engineered the glycogen metabolism of MSCs to promote glycogen accumulation. We tested several engineering strategies in human HEK 293T cells and primary MSCs derived from mice. Overexpression of phosphorylation-resistant GYS (GYSmut), as well as the co-expression of GYS and UGP, induced intense glycogen. Glycogen engineering enhanced the starvation resistance of MSCs and improved their viability post-implantation. Compared with the control group, the engineered MSCs exhibited a significantly enhanced therapeutic effect on PF, indicating the importance of glucose metabolism regulation for MSC-based therapy.

Glycogen-engineered MSCs may serve as chassis cells for further applications by enhancing cell viability post-implantation, and our glycogen engineering strategies also have the potential to be applied to other therapeutic cells. Transient expression of GYSmut by mRNA transfection may be a safer alternative than lentiviral transduction, avoiding the risks of chromosomal modification. LNP-mediated circular mRNA transfection would further improve engineering efficiency (*Huang et al., 2024*).

We investigated the effect of glycogen engineering on the transcriptome of MSCs by RNA-seq, which revealed large-scale gene transcription changes. In future studies, complex regulatory networks may be elucidated at the levels of proteomics and metabolomics. Previous studies have revealed the role of glucose metabolism regulation in MSCs' immunoregulatory properties (*Contreras-Lopez et al., 2020*; *Luo et al., 2023*). The broader impact of glycogen engineering on MSCs remains to be elucidated. We tested the therapeutic efficacy of glycogen-engineered MSCs in the bleomycin-induced PF mice model. Their efficacy in other disease models is still to be investigated.

Within the context of MSC-based cellular therapies, administered cells were distributed to various microenvironments characterized by variable oxygen concentration, nutrient availability, and immune responses. While in vivo animal models offer a more physiologically relevant platform for validation, they are less convenient for detailed investigation due to multifactorial influences, making it challenging to fully understand the precise kinetic properties and adaptation dynamics of the implanted cells, which is a limitation of this study.

In previous studies, complex gene circuits were introduced to enhance the efficacy of various cell-based therapies, including MSCs and CAR-T cells (*Allen et al., 2022*; *Cheng et al., 2019*), and some research also revealed the advantages of metabolic engineering (ME) (*Ye et al., 2022*). However, in contrast to extensive ME research in prokaryotic organisms, ME of mammalian therapeutic cells has yet to be developed. Here, we demonstrated the great potential of glycogen engineering, offering a basis for future studies. By dynamically controlling essential enzymes, cellular metabolism can be remodeled to achieve optimal survival and therapeutic efficacy post-implantation. Further research in this area will provide additional opportunities for cell therapy, gene therapy, and tissue engineering.

## Materials and methods

**Key resources table**

| Reagent type (species) or resource | Designation | Source or reference | Identifiers | Additional information |
|---|---|---|---|---|
| Gene (*Mus musculus*) | *Gys1* | GenBank | NM_030678 | |
| Gene (*M. musculus*) | *Gbe1* | GenBank | NM_028803.4 | |
| Gene (*M. musculus*) | *Gyg1* | GenBank | NM_001355261 | |

*Continued on next page*

*Continued*

| Reagent type (species) or resource | Designation | Source or reference | Identifiers | Additional information |
| --- | --- | --- | --- | --- |
| Gene (*M. musculus*) | *Ugp2* | GenBank | NM_001290634.1 | |
| Strain, strain background (*Escherichia coli*) | DH5a | Vazyme | Cat#C502-02 | Competent cell |
| Cell line (*M. musculus*) | MSC | This paper | | Primary MSCs isolated from adipose tissue of C57/B6J mice (male) |
| Cell line (*Homo sapiens*) | HEK293T | ATCC | RRID:CVCL_0063 | |
| Transfected construct (*M. musculus*) | pCDH | Addgene | RRID:Addgene_72266 | Lentiviral construct to transfect and express GYS/GYG/GBE/UGP, etc. |
| Sequence-based reagent | Lpl-F | Synthesized by Tsingke, Beijing | PCR primers | TTGCCCTAAGGACCCCTGAA |
| Sequence-based reagent | Lpl-R | Synthesized by Tsingke, Beijing | PCR primers | TTGAAGTGGCAGTTAGACACAG |
| Sequence-based reagent | Actb-F | Synthesized by Tsingke, Beijing | PCR primers | CCACTGTCGAGTCGCGTCC |
| Sequence-based reagent | Actb-R | Synthesized by Tsingke, Beijing | PCR primers | TCCGGAGTCCATCACAATGC |
| Commercial assay or kit | Lipo8000 | Beyotime | Cat#C0533 | |
| Commercial assay or kit | Glucogen Content Assay Kit | SOLARBIO | BC0345-100T | |
| Commercial assay or kit | PAS staining kit | Beyotime | Cat#C0142S | |
| Commercial assay or kit | Lentivirus Concentration Solution | Servicebio | Cat#G1801-100ML | |
| Commercial assay or kit | CCK-8 assay kit | BIORIGIN | Cat#BN15201 | |
| Commercial assay or kit | Oil Red O staining kit | Beyotime | Cat#C0157S | |
| Commercial assay or kit | Secrete-Pair Dual Luminescence Assay Kit | GeneCopoeia | Cat#LF061 | |
| Commercial assay or kit | RNA isolator Total RNA Extraction Reagent | Vazyme | Cat#R401-01 | |
| Chemical compound, drug | Akalumine-HCL | InvivoChem | Cat#V41343 | |
| Chemical compound, drug | bleomycin | Sigma Aldrich | Cat#B5507-15un | |

## Animals and experimental design

All the animal experiments were authorized by the institutional animal care and use committee of Tsinghua University. Six-week-old male C57BL/6J mice were housed in the specific pathogen-free experimental animal environment at the laboratory animal research center of Tsinghua University, with 5–6 mice per cage having access to sterilized food and water ad libitum under a 12 hr light/dark cycle. Mice were anesthetized with avertin (200 µL per mouse) or 2.5% tribromoethanol. And euthanized by cervical dislocation or $CO_2$ inhalation. 76 mice were used in total. To obtain reliable results, each group contains at least three mice. In the same experiment, the mice in each group came from the same batch and were randomly captured, processed, and assigned to different cages. The number of mice is randomly assigned, the surgeon knows the grouping situation, and the statistical analyst does not know the grouping in advance. All animal experiments were authorized by the institutional animal care and use committee of Tsinghua University (Approval No. 23-WQ1.G23-1, Study on different factors on the survivability of implanted MSCs in murine pulmonary fibrosis model, 2023-01-18 to 2026-04-12).

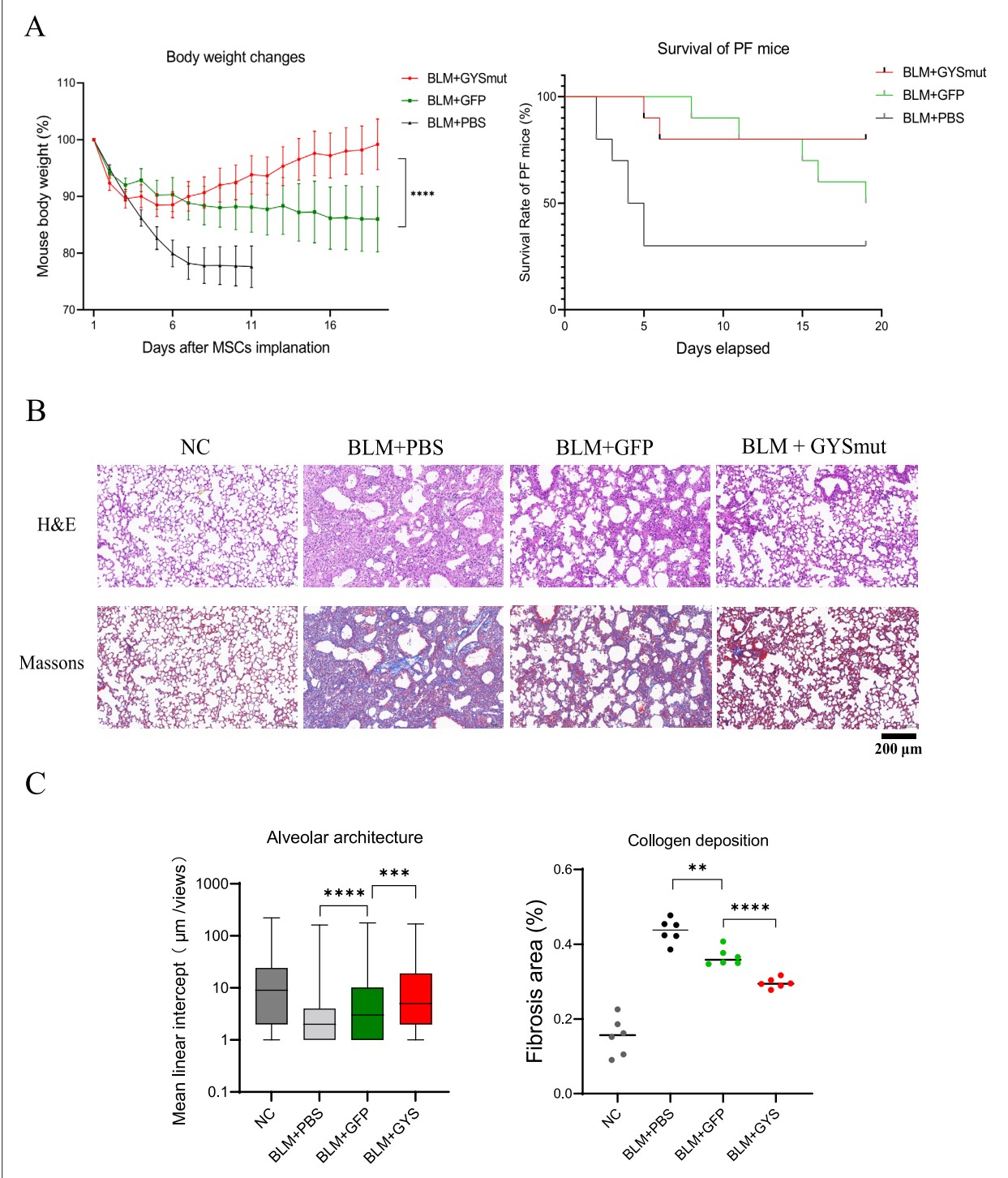

**Figure 5.** Therapeutic efficacy of glycogen-engineered mesenchymal stem cells (MSCs). (**A**) Survival and body weight changes of pulmonary fibrosis (PF) mice treated with GYSmut MSCs and control cells (10 mice each group, mean ± SEM, two-way ANOVA p-value = 0.0001). (**B**) Representative lung tissue sections stained with hematoxylin and eosin (H&E) and Masson's trichrome. NC group is healthy mice.Scale bar, 200 μm (**C**) Collagen deposition and preserved alveolar size (quantified by mean linear intercept [MLI]) of lung tissue sections (six mice each group).

## Isolation of mouse adipose-derived stem cells

MSCs were obtained by euthanizing the mice, digesting the inguinal adipose tissue with 1.0 mg/mL collagenase type I (Sigma, SCR103), and culturing the released cells in DMEM/F-12 medium (Hyclone, SH30023.FS) in a humidified atmosphere containing 5% carbon dioxide at 37°C, according to a previously published method (*Rahman et al., 2022*).

## Construction of the BLM-induced PF mouse model and administration of MSCs

Mice were anesthetized with 2.5% tribromoethanol (0.8% NaCl, 1 mM Tris [pH 7.4], 0.25 mM EDTA [pH 7.4]), and administered 3.5 mg/kg bleomycin solution (Sigma, B5507-15un) in 50 µL phosphate-buffered saline (PBS) intratracheally on day 0 using a 1 mL syringe with a 25G needle. Control animals were mock-treated with PBS (*Rahman et al., 2022*).

Injection of MSCs (at passages 3–6) was performed on day 3 after BLM treatment. Each animal received an intratracheal injection of 50 µL of a suspension comprising $1\times10^7$ cells/mL in PBS ($5\times10^5$ cells/mouse) using a 1 mL disposable syringe with a 25G needle. For the control and BLM+PBS groups, 50 µL PBS was injected instead. Before implantation, the cells were washed three times to remove the culture medium. The body weight of treated mice was recorded daily (*Rahman et al., 2022*; *Tzouvelekis et al., 2011*).

## Histological evaluation of lung damage and collagen deposition of BLM-induced lung injury

To assess the development of PF and therapeutic efficacy, mice were evaluated on day 7. For lung tissue collection, mice were euthanized by $CO_2$ inhalation. The thoracic cavity was opened, and the lungs were perfused and washed with ice-cold PBS. The lungs were then fixed with 4% (vol/vol) paraformaldehyde in PBS and processed into 5-µm-thick paraffin sections, which were mounted on glass slides.

For histological assessment of alveolar architecture and fibrosis, sections were stained with H&E or Masson's trichrome stain, respectively. The samples were scanned using Pannoramic SCAN (3DHIS-TECH) with a 20× objective. Fibrosis area percentages and alveolar size (quantified by MLI) were calculated according to a previously published method (*Rahman et al., 2022*; *Crowley et al., 2019*).

## Generation of genetically engineered MSCs and in vitro tests

HEK293T cells were cultured in DMEM with 10% FBS. Sequences of essential enzymes are amplified from cDNA of mouse MSCs. To test combinations of essential enzymes, HEK293T cells in six-well plates were transiently transfected with encoding plasmids (2 µg/well) using Lipo8000 (Beyotime, C0533). Co-expressed enzymes were connected by the 2A sequence. After 3 days, the glycogen content of each group was quantified using a commercial kit (SOLARBIO, BC0345-100T). A PAS staining kit (Beyotime, C0142S) was used to stain glycogen granules.

To construct the lentiviral vectors, designed gene sequences were cloned into the pCDH backbone and used to co-transfect HEK293T cells with pMD2.G and psPAX2 plasmids using Lipo8000. Essential enzymes were co-expressed by connecting them with EGFP or puromycin N-acetyltransferase via the 2A sequence. The cell supernatant was collected every 24 hr for 3 days. The viral particles were concentrated using Lentivirus Concentration Solution (Servicebio, G1801-100ML), after which the infectious solution was prepared by resuspending them in 200 µL PBS and stored at –80°C until use. When the density of MSCs in the six-well plate reached approximately 80%, 20 µL of the lentiviral solution was added. Following 2 days after infection, puromycin was added to a final concentration of 1 mg/L for selection.

Glycogen was quantified after 3 days of induction. To simulate starvation conditions, DPBS solution was used to replace the culture medium of seeded MSCs (96-well plate, 5000 cells/well), and a CCK-8 assay kit (BIORIGIN, BN15201) was used to measure cell viability. To test the adipogenic differentiation potential of MSCs, we induced them with commercial adipocyte medium (Procell, PD-027) for 5 days as described in the manufacturer's protocol, and then performed Oil Red O staining (Beyotime, C0157S). qRT-PCR of gene Lpl was performed, using primers Lpl-F and Lpl-R, and gene Actb was used as reference gene, using primers Actb-F and Actb-R.

## Viability detection and live imaging of MSCs post-implantation in the PF model

To detect viability changes of MSCs post-implantation, 2.5×10⁵ MSCs expressing *Gaussia* luciferase suspended in 50 µL of PBS were injected intratracheally into PF mice, as described above. The whole lungs of mice were collected on the day of surgery (D0) and the 7th day (D7) after surgery (n=3 each time point) and were stored in a 1.5 mL centrifuge tube at –80°C. Then, 400 µL PBS and twenty 1-mm beads (Servicebio, G0201) were added to the centrifuge tube with lungs for processing with a KZ-III high-speed homogenizer (Servicebio) at 60 Hz for 30 s, six times. The homogenate was centrifuged (10,000×*g*, 5 min), and the activity of *Gaussia* luciferase in the suspension was detected using a commercial kit (GeneCopoeia, LF061). The lungs of cells without *Gaussia* luciferase expression were used as negative controls, and the corresponding signals were considered to represent background subtraction in the calculations. Cell viability changes of each group were assessed by calculating the ratio of the intensity of lung tissue collected on D7 to D0.

To detect the in vivo distribution of MSCs post-implantation, MSCs expressing Akaluc were injected intratracheally into PF mice (n=5), as described above. Live imaging was performed on the 3rd and 7th days after implantation. Akalumine-HCL (InvivoChem, V41343) was dissolved in PBS (2.5 mg/mL) and intraperitoneally injected into mice (100 µL/mouse). Luminescence was detected after 10 min using a Lumina III instrument and analyzed with Living Image software version 4.8.

## RNA-seq of MSCs

MSCs were cultured in 100 mm dishes (5×10⁶ cells per dish) in DMEM/F-12 medium. Trizol (Vazyme, R401-01) was used for RNA extraction. Illumina RNA-seq and subsequent analysis were performed by ANNOROAD Corporation (Beijing, China). DESeq2 was used to identify differentially expressed genes (|log2 fold-change|≥1, adjusted p-value<0.05).

## Cell lines

HEK293T cell line was from ATCC, of which STR profiling was confirmed. Test of mycoplasma contamination showed negative.

## Statistical analysis

Statistical analysis was performed using GraphPad Prism software version XY (GraphPad Software Inc). Significant differences between groups were identified using unpaired Student's *t*-test or two-way ANOVA. Asterisks were used to indicate significance in figures (*: p<0.05, **: p<0.01, ***: p<0.001, ****: p<0.0001). Unless indicated otherwise, all quantitative data are presented as mean values ± SD. All statistical data shown in figures describe biological replicates.

# Acknowledgements

We thank the team of Du Yanan at Tsinghua University for sharing the plasmid encoding Akaluc. We also thank Prof. George Guo-Qiang Chen at Tsinghua University for his advice on this project. The authors declare that they have not used Artificial Intelligence in this study. This work was supported by the Ministry of Science and Technology of China (No. 2021YFC2101700 and 2018YFA0900100 to QW), the Natural Science Foundation of China (No. 31961133019 to QW) and Vanke Special Fund for Public Health and Health Discipline Development, Tsinghua University (2022Z82WKJ006).

# Additional information

### Funding

| Funder | Grant reference number | Author |
| --- | --- | --- |
| Ministry of science and technology of China | 2021YFC2101700 | Qiong Wu |
| Ministry of science and technology of China | 2018YFA0900100 | Qiong Wu |

| Funder | Grant reference number | Author |
| --- | --- | --- |
| The Natural Science Foundation of China | 31961133019 | Qiong Wu |
| Public Health and Health Discipline Development, Tsinghua University | 2022Z82WKJ006 | Qiong Wu |

The funders had no role in study design, data collection and interpretation, or the decision to submit the work for publication.

## Author contributions

Yongyue Xu, Conceptualization, Data curation, Validation, Investigation, Visualization, Methodology, Writing – original draft, Project administration, Writing – review and editing; Mamatali Rahman, Formal analysis, Validation, Investigation, Methodology, Writing – original draft; Zhaoyan Wang, Bo Zhang, Hanqi Xie, Lei Wang, Haowei Xu, Investigation, Methodology; Xiaodan Sun, Shan Cheng, Writing – review and editing; Qiong Wu, Conceptualization, Supervision, Funding acquisition, Investigation, Project administration

## Author ORCIDs

Yongyue Xu ![ORCID] https://orcid.org/0009-0005-8239-8968
Mamatali Rahman ![ORCID] https://orcid.org/0000-0002-4321-0599
Qiong Wu ![ORCID] https://orcid.org/0000-0002-9048-9094

## Ethics

All animal experiments were authorized by the institutional animal care and use committee of Tsinghua University (Approval No. 23-WQ1.G23-1, Study on different factors on the survivability of implanted MSCs in murine pulmonary fibrosis model, 2023-01-18 to 2026-04-12). All surgery was performed under avertin anesthesia, and every effort was made to minimize suffering.

Reviewer #1 (Public review): https://doi.org/10.7554/eLife.106023.3.sa1
Reviewer #2 (Public review): https://doi.org/10.7554/eLife.106023.3.sa2
Author response https://doi.org/10.7554/eLife.106023.3.sa3

# Additional files

## Supplementary files

Supplementary file 1. Protein sequences used in this study.

MDAR checklist

## Data availability

The dataset generated by this study is available in GSE281974 [https://www.ncbi.nlm.nih.gov/geo/query/acc.cgi?acc=GSE281974].

The following dataset was generated:

| Author(s) | Year | Dataset title | Dataset URL | Database and Identifier |
| --- | --- | --- | --- | --- |
| Xu Y, Rahman M, Wang Z, Zhang B, Xie H, Wang L, Xu H, Sun X, Cheng S, Wu Q | 2025 | The effect of expressing dephosphorylated GYS on MSCs | https://www.ncbi.nlm.nih.gov/geo/query/acc.cgi?acc=GSE281974 | NCBI Gene Expression Omnibus, GSE281974 |

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
