## [Editor Report · eLife Assessment]

This **important** study presents a novel approach to enhance the therapeutic potential of mesenchymal stromal cells (MSCs) by genetically modifying their glycogen synthesis pathway, resulting in increased glycogen accumulation and improved cell survival under starvation conditions, particularly in the context of experimental pulmonary fibrosis. The methods and findings are generally **solid** and could be strengthened in the future by investigating the kinetics of persistence, the immunomodulatory effects, and the underlying improved mechanism of action of MSCs in this pulmonary fibrosis model. If confirmed, this approach could suggest potential methods to improve the therapeutic functionality of MSCs in cell therapy strategies.

---

## [Referee Report · Reviewer #1 (Public review)]

Summary:

This study provides the first evidence that glucose availability, previously shown to support cell survival in other models, is also a key determinant for post-implantation MSC survival in the specific context of pulmonary fibrosis. To address glucose depletion in this context, the authors propose an original, elegant, and rational strategy: enhancing intracellular glycogen stores to provide transplanted MSCs with an internal energy reserve. This approach aims to prolong their viability and therapeutic functionality after implantation.

Strengths:

The efficacy of this metabolic engineering strategy is robustly demonstrated both in vitro and in an orthotopic mouse model of pulmonary fibrosis.

---

## [Referee Report · Reviewer #2 (Public review)]

Summary:

In this article, the authors investigate enhancing the therapeutic and regenerative properties of mesenchymal stem cells (MSCs) through genetic modification, specifically by overexpressing genes involved in the glycogen synthesis pathway. By creating a non-phosphorylatable mutant form of glycogen synthase (GYSmut), the authors successfully increased glycogen accumulation in MSCs, leading to significantly improved cell survival under starvation conditions. The study highlights the potential of glycogen engineering to improve MSC function, especially in inflammatory or energy-deficient environments. However, critical gaps in the study's design, including the lack of validation of key findings, limited differentiation assessments, and missing data on MSC-GYSmut resistance to reactive oxygen species (ROS), necessitate further exploration.

Strengths:

(1) Novel Approach: The study introduces an innovative method of enhancing MSC function by manipulating glycogen metabolism.

(2) Increased Glycogen Storage: The genetic modification of GYS1, resulting in GYSmut, significantly increased glycogen accumulation, leading to improved MSC survival under starvation, which has strong implications for enhancing MSC therapeutic properties in energy-deficient environments.

(3) Potential Therapeutic Impact: The findings suggest significant therapeutic potential for MSCs in conditions that require improved survival, persistence, and immunomodulation, especially in inflammatory or energy-limited settings.

(4) In Vivo Validation: The in vivo murine model of pulmonary fibrosis demonstrated the improved survival and persistence of MSC-GYSmut, supporting the translational potential of the approach.

Weaknesses:

(1) Lack of Differentiation Assessments: The study did not evaluate key MSC differentiation pathways, including chondrogenic and osteogenic differentiation. The absence of analysis of classical MSC surface markers and multipotency limits the understanding of the full potential of MSC-GYSmut.

(2) Missing Validation of RNA Sequencing Data: Although RNA sequencing data revealed promising transcriptomic changes in chondrogenesis and metabolic pathways, these findings were not experimentally validated, limiting confidence.

(3) Lack of ROS Resistance Analysis: Resistance to reactive oxygen species (ROS), an important feature for MSCs under regenerative conditions, was not assessed, leaving out a critical aspect of MSC function.

(4) Limited Exploration of Immunosuppressive Properties: The study did not address the immunosuppressive functions of MSC-GYSmut, which are critical for MSC-based therapies in clinical settings.

Conclusion:

The study presents an exciting new direction for enhancing MSC function through glycogen metabolism engineering. While the results show promise, key experiments and validations are missing, and several areas, such as differentiation capacity, ROS resistance, and immunosuppressive properties, require further investigation. Addressing these gaps would solidify the conclusions and strengthen the potential clinical applications of MSC-GYSmut in regenerative medicine.

---

## [Author Response]

The following is the authors’ response to the original reviews.

**Reviewer #1 (Public Review)**：(1) Glycogen biosynthesis typically involves several enzymes. In this context, could the authors comment on the effect of overexpressing a single enzyme - especially a mutant version - on the structure or quality of the glycogen synthesized?

While quantitative molecular weight analysis of synthesized glycogen was not performed, we documented changes in glycogen particle morphology. GYSmut overexpression resulted in significantly enlarged singular glycogen granules, suggesting potential high molecular mass, while GYS-GYG co-overexpression in MSCs (GYG being the essential enzyme for glycogen synthesis initiation) produced a diffuse glycogen distribution pattern rather than particulate structures. We have incorporated this result as new Figure S2C.

These results suggest that overexpression of specific glycogen-metabolizing enzymes significantly influences glycogen structure. Consequently, targeted modulation of glycogen architecture and properties through key enzymes represents a potential avenue for future investigation.

(2) Regarding the in vitro starvation experiments (Figure 2C), what oxygen conditions (pO₂) were used? Are these conditions physiologically relevant and representative of the in vivo lung microenvironment?

Our in vitro starvation experiments (Figure 3C) were conducted under normoxic (21%). The oxygen concentration in human lungs is physiologically lower than atmospheric levels, with healthy individuals exhaling air containing approximately 16% oxygen (Thalakkotur Lazar Mathew, Diagnostics 2015). To our knowledge, direct measurements of alveolar oxygen concentration in pulmonary fibrosis are rare. Therefore, to evaluate the performance of GYSmut under hypoxic conditions, in the revised manuscript, Figure S2 has been augmented to include assessment of cell performance under combined hypoxia （oxygen concentration < 5%）and nutrient deprivation stress, which further corroborate the superiority of the GYSmut group over the control under different oxygen concentrations.

(3) In the in vitro model, how many hours does it take for the intracellular glycogen reserve to be completely depleted under starvation conditions?

While quantitative cell viability data were recorded up to 72 hours post-implantation (Fig 3C), we observed cell viability at approximately 96 hours. We noticed that the presence of glycogen particles exhibited a correlation with sustained cell viability. However, reliable quantitative assessment of glycogen became increasingly challenging upon significant depletion of viable cells, thereby limiting our measurements during later time points.

(4) For the in vivo model, is there a quantitative analysis of the survival kinetics of the transplanted cells over time for each group? This would help to better assess the role and duration of glycogen stores as an energy buffer after implantation.

We tracked the in vivo distribution and persistence of implanted MSCs using enzymatic activity quantification assays (using Gluc luciferase assay) and live animal imaging (using Akaluc luciferase). The revised manuscript includes quantitative analysis of the in vivo fluorescence imaging data, which has been supplemented as Figure S4. Glycogen-engineered MSCs and control cells were quantitatively assessed at three discrete time points post-implantation. This quantification revealed a transient divergence in cell viability between the experimental and control groups around day 7. However, fluorescence in both cohorts subsequently declined to similar levels over the extended observation period.

(5) Finally, the study was performed in male mice only. Could sex differences exist in the efficacy or metabolism of the engineered MSCs? It would be helpful to discuss whether the approach could be expected to be similarly effective in female subjects.

We appreciate the reviewer’s important question regarding potential sex differences. Our study used male mice based on three key considerations: (1) Clinical Relevance: Idiopathic pulmonary fibrosis (IPF) shows significant male predominance, with diagnosis rates 3.5-fold higher in men (37.8% vs 10.6%, p<0.0001) and greater diagnostic confidence (Assayag et al., Thorax 2020). (2) Model Consistency: The bleomycin model (our chosen method) demonstrates more consistent fibrotic responses in male mice (Gul et al., BMC Pulm Med 2023). (3) Biological Rationale:

Estrogen’s protective effects in females may confound therapeutic assessments (cited in Assayag et al.).

We fully acknowledge this limitation and will include female subjects in subsequent translational studies. The therapeutic principle should theoretically apply to both sexes, but we agree this requires experimental validation.

(6) The number of mice for each group and time point should be specified.

The manuscript text has been revised to enhance clarity, and the number of mice for each group and time point has been specified (line 170 to 182).

**Reviewer #2 (Public Review):**
(4) Inconsistencies in In Vivo Data: There is a discrepancy between the number of animals shown in the figures and the graph (three individuals vs. five animals), as well as missing details on how luciferase signal intensity was quantified, requiring further clarification.

To assess MSC survival in vivo, we employed two strategies utilizing distinct luciferases optimized for specific detection modalities. MSC viability was quantified ex vivo through Gaussia luciferase (Gluc) activity, leveraging its high sensitivity and established commercial assay kits (n = 3 mice per group per time point). For non-invasive longitudinal tracking within living animals, MSC distribution and viability were monitored via in vivo bioluminescence imaging using Akaluc luciferase, selected for its superior tissue penetration and sensitivity in situ (n = 5 mice per group).The manuscript text has been revised to enhance clarity, and the experiment protocols for luciferase signal detection and quantification has been added into Methods.

（1) (2) (3) (5):

We fully agree that further investigation into the functional consequences of glycogen engineering in MSCs – encompassing core cellular functions, immunomodulatory properties, and associated signaling pathways – is important to fully elucidate the underlying mechanisms. Cellular metabolism is intrinsically intertwined with diverse physiological processes. Consequently, we believe that glycogen engineering exerts multifaceted effects on MSCs, likely extending beyond the modulation of any single specific pathway. Studying the metabolic perturbation induced by such engineering approaches in mammalian cells represents an interesting field. The exploration of these aspects remains an long-term research objective within our group.

**Reviewer #2 (Recommendations for the authors):**
(6) Clarification of Data in the Murine Model:In Figure 4B, there is a discrepancy between the number of animals shown in the image (five) and those represented in the graph (three). This discrepancy needs clarification. Additionally, the study lacks information regarding the intensity of the signal in the luciferase assays. It is unclear how luciferase expression in the mice was quantified, and providing this detail would enhance the understanding of the data presented.

We sincerely appreciate these valuable suggestions. We have revised the relevant text for greater clarity. Figure 4B and Figure 4C present results from two distinct experimental approaches, each employing different luciferase reporters and measurement methodologies, and different num of mice were used in these two experiments.

Quantitative data derived from the in vivo bioluminescence imaging has been supplemented as Figure S4. The experiment protocols for luciferase signal detection and quantification has been added into Methods.

To other recommendations of reviewer 2：

We sincerely appreciate your valuable insights, which demonstrate your deep expertise. We fully agree that beyond nutrient availability, factors such as reactive oxygen species (ROS) and the immune microenvironment are also critical limitations affecting the survival and therapeutic efficacy of implanted MSCs.

We propose that glycogen engineering exerts broad effects on MSCs. These effects manifest as changes in multiple cellular characteristics, including proliferation, differentiation, surface marker expression, antioxidant capacity, and immunomodulatory activity – all crucial factors for the therapeutic purpose of MSCs.

We believe these changes likely involve complex networks of interconnected regulatory factors. The underlying mechanisms might be clarified through proteomic and metabolomic profiling.

However, comprehensively investigating these interconnected aspects requires significant time and resources. Some components of this research extend beyond the current scope of our project. Nevertheless, exploring these mechanisms remains an important objective, and we will actively work to investigate them further in our ongoing studies.